# Periodontitis promotes bacterial extracellular vesicle-induced neuroinflammation in the brain and trigeminal ganglion

Jae Yeong Ha[1,2☯], Jiwon Seok[3☯], Suk-Jeong Kim[1,2], Hye-Jin Jung[3], Ka-Young Ryu[2,3], Michiko Nakamura[4,5], Il-Sung Jang[4,5], Su-Hyung Hong[1], Youngkyun Lee[2,3]*, Heon-Jin Lee[1,2]*

1 Department of Microbiology and Immunology, School of Dentistry, Kyungpook National University, Daegu, Korea, 2 Craniofacial Nerve-Bone Network Research Center, Kyungpook National University, Daegu, Korea, 3 Department of Biochemistry, School of Dentistry, Kyungpook National University, Daegu, Korea, 4 Department of Pharmacology, School of Dentistry, Kyungpook National University, Daegu, Korea, 5 Brain Science & Engineering Institute, Kyungpook National University, Daegu, Korea

☯ These authors contributed equally to this work.
* ylee@knu.ac.kr (YL); heonlee@knu.ac.kr (H-JL)

**Data Availability Statement:** All relevant data are within the manuscript and its Supporting Information file.

## Abstract

Gram-negative bacteria derived extracellular vesicles (EVs), also known as outer membrane vesicles, have attracted significant attention due to their pathogenic roles in various inflammatory diseases. We recently demonstrated that EVs secreted by the periodontopathogen *Aggregatibacter actinomycetemcomitans* (Aa) can cross the blood–brain barrier (BBB) and that their extracellular RNA cargo can promote the secretion of proinflammatory cytokines, such as IL-6 and TNF-α, in the brain. To gain more insight into the relationship between periodontal disease (PD) and neuroinflammatory diseases, we investigated the effect of Aa EVs in a mouse model of ligature-induced PD. When EVs were administered through intragingival injection or EV-soaked gel, proinflammatory cytokines were strongly induced in the brains of PD mice. The use of TLR (Toll-like receptor)-reporter cell lines and *MyD88* knockout mice confirmed that the increased release of cytokines was triggered by Aa EVs via TLR4 and TLR8 signaling pathways and their downstream MyD88 pathway. Furthermore, the injection of EVs through the epidermis and gingiva resulted in the direct retrograde transfer of Aa EVs from axon terminals to the cell bodies of trigeminal ganglion (TG) neurons and the subsequent activation of TG neurons. We also found that the Aa EVs changed the action potential of TG neurons. These findings suggest that EVs derived from periodontopathogens such as Aa might be involved in pathogenic pathways for neuroinflammatory diseases, neuropathic pain, and other systemic inflammatory symptoms as a comorbidity of periodontitis.

## Author summary

Extracellular vesicles (EVs) secreted from bacterial cells play a key role in microbe-host cell communication. Bacterial EVs (bEVs) may be closely linked to the pathogenesis

**Funding:** This research was supported by the Basic Science Research Program through the National Research Foundation of Korea (NRF) and funded by the Korean Government (MSIT 2017R1A5A2015391) to YL and H-JL, (2021R1A2C1004512) to H-JL, and (2020R1A2C1009364) to YL. The funder had no role in study design, data collection and analysis, decision to publish, or preparation of the manuscript.

**Competing interests:** The authors have declared that no competing interests exist.

underlying neuroinflammatory diseases. In the current experiments, bEVs caused the brain to release proinflammatory cytokines by activating the host TLR signaling pathway. This induction of neuroinflammation was significantly more prominent in the periodontitis disease model, implying a substantial link between periodontal and neuroinflammatory diseases through bEVs. We also show that bEVs are translocated to the neuronal cell body via retrograde axonal transport, where they directly activate neuronal proinflammatory signals. Our findings reveal that bEVs are a pathogenic pathway for neurological conditions potentially linked to periodontitis, such as Alzheimer's disease and trigeminal neuropathic pain.

## Introduction

Periodontal disease (PD) is a common infectious disease affecting people worldwide. It develops as a result of the inflammatory response of periodontium to periodontopathogenic bacteria and their products [1]. Numerous studies have reported that inflammation in PD could be the initial molecular pathology prior to neurodegeneration onset [2]. In particular, epidemiological studies have postulated a relationship between periodontitis and Alzheimer's disease (AD) [3,4]. Recently, periodontopathogen *Porphyromonas gingivalis* was found in the brain of AD patients [5], suggesting that oral microorganisms may colonize the brain and induce neurodegenerative diseases. In addition, patients with chronic periodontitis show an elevated risk of AD or vascular dementia [6]. Among the major PD pathogens, *Aggregatibacter actinomycetemcomitans* (Aa) is a small gram-negative bacterium associated with localized aggressive periodontitis [7]. Through the activation of inflammatory pathways, Aa is believed to be involved in a number of systemic disorders including endocarditis, rheumatoid arthritis, and osteoporosis [8,9].

Bacterial extracellular vesicles (bEVs) are membrane vesicles released from both gram-positive and gram-negative bacteria, with sizes ranging from 20 to 400 nm. bEVs were first discovered in *Escherichia coli* and were therefore named outer membrane vesicles due to the specific cell wall component of gram-negative bacteria, the outer membrane [10]. Almost all cell types of life domains secrete these membrane vesicles, collectively termed extracellular vesicles (EVs). Although the initially suggested role of bEVs was the disposal of cellular garbage, current evidence indicates that EVs carry functional molecules as their cargos for cell-to-cell communication and play an even bigger role in microbe-to-host communication [11,12]. bEVs are loaded with various bioactive molecules such as proteins, lipids, DNA, and RNA [13,14]. These bacteria-originated components can be delivered by EVs to other cells via endocytosis, lipid rafts, and/or membrane fusion [13,15,16]. Among the numerous macromolecules that can be transmitted by bEVs, EV RNAs (extracellular RNAs or exRNAs) have garnered special attention due to their biological activity as microbe-to-host communication molecules [17,18]. Furthermore, bEVs in the systemic circulation are potential causes of several human diseases, including inflammatory bowel disease, cancer, depression, and liver disorders [19]. Previously we showed that EVs of Aa and their RNA cargo could cross the blood-brain barrier (BBB) and increase proinflammatory cytokine expression in the mouse brain [20,21]. This was the first direct evidence that blood-borne bEVs can cross the mouse BBB and cause neuroinflammation. However, it remained unclear whether EVs originated from periodontopathogens can enter the brain through leaky gum in humans with PD.

In this study, we examined the role of periodontal disease in the migration of Aa EVs into brain and trigeminal ganglion (TG) and subsequent neuroinflammation. Furthermore, the excitability of TG neurons isolated from the mice challenged with Aa EVs was evaluated using

an electrophysiology test. Our findings not only provide fresh insight into the pathogenicity of bEV-induced trigeminal neuralgia, but also propose a novel pathway implicating bEVs in the development of neuroinflammatory diseases including Alzheimer's and Parkinson's as a distal consequence of periodontitis.

## Results

### Basic characterization of Aa EVs

From an Aa bacterial culture with a concentration of $5.66 \times 10^9$ CFU/ml, we successfully purified approximately $4.5 \times 10^{10}$ Aa EV particles/ml. This indicates that about 7.9 EV particles were produced from each bacterial cell. In this study, we administered a dose of $2.25 \times 10^9$ Aa EV particles in a 50 μl injection or gel. The size distribution of Aa EVs utilized in this study showed that the majority of EVs exhibited an average diameter of 104.1 nm (Fig 1A). The structural integrity of Aa EVs was confirmed through transmission electron microscopy

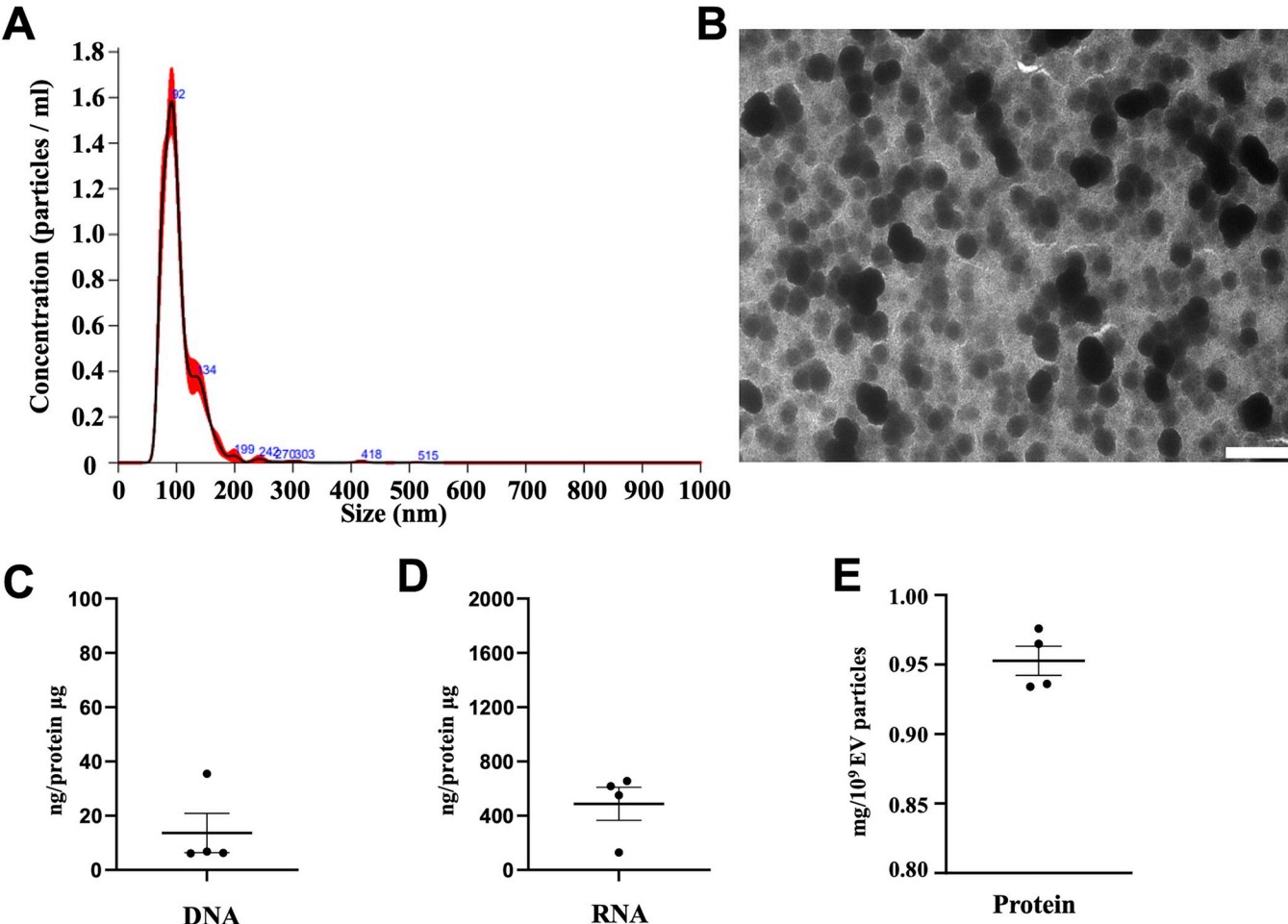

**Fig 1. Purification and analysis of Aa EVs.** (A) Size distribution (X-axis) and particle numbers (Y-axis) of Aa EVs analyzed using the NanoSight system. (B) Transmission electron micrograph of Aa EVs, displaying small vesicles ranging from 50 to 200 nm in diameter. Original magnification: ×6,000; scale bar: 500 nm. DNA (C), RNA (D), and protein (E) content of Aa EVs are presented. Aa EVs were purified using a purification kit, with RNase A or DNase I employed to eliminate free nucleic acids in EV samples. Following EV lysis, 1 ml of RNase A or DNase I was added to 1 ml of the lysed EV and incubated at 37°C for 25 min to obtain individual EV RNA cargo. EV cargo DNA or RNA was measured and normalized to 1 μg of protein. Data represent individual biological EV samples and display the mean ± SEM of four biological replicates.

(Fig 1B). We quantified the amounts of DNA, RNA, and protein associated with Aa EVs. In order to accurately measure the cargo content within the particles, free nucleic acids located outside the vesicles were removed via an initial nuclease treatment. Aa EVs were found to contain significantly more RNA than DNA (Fig 1C and 1D). The average protein content in Aa EVs was determined to be 0.953 mg per $10^9$ EV particles (Fig 1E).

## Aa EVs induce TNF-α and IL-6 expression in the brain of mice with periodontitis ligature

Prior to this study, we demonstrated that intracardiac injection of Aa EVs in mice resulted in successful delivery to the brain after crossing the BBB, and that exRNA cargos promoted TNF (tumor necrosis factor)-α expression in the mouse brain [21]. In addition, we also demonstrated that intravenous Aa EVs and their exRNA cargo were sequentially delivered into meningeal macrophages and cortex microglial cells [20]. The spread of Aa EVs to systemic organs after intravenous tail vein injection was demonstrated in S1 Fig. Here, we postulated that the migration of local (oral) bacterial EVs to the brain might be facilitated when the physical barriers are compromised by periodontal diseases. In Fig 2A, periodontitis was induced by placing silk ligature around the maxillary left 2nd molar. The efficacy of ligature and subsequent tissue damage was confirmed by histological staining, tartrate-resistant acid phosphatase (TRAP), since the periodontal inflammation by ligature was proven to induce osteoclastogenesis and bone resorption [22]. The number of osteoclasts in alveolar bones was significantly elevated as early as at 24 h after ligation compared with control (Fig 2B). We first speculated how Aa EV injected intragingivally in the PD mice affects the brain. Fluorescence intensity in the brain measured by IVIS (In Vivo Imaging System) showed elevated Aa EV signals in ligature-induced PD mice compared to sham controls (Fig 2C). Aa EVs appeared more prominently in the PD mice compared to sham control mice after 24 h of intragingival injection (Fig 2D). Both TNF-α and IL (interleukin) -6 were dramatically upregulated in immunohistochemistry conducted from confocal microscopy (Figs 2E and S2) and ELISA (Fig 2F and 2G) compared to non-EV injected and sham-operated control mice.

To further corroborate the facilitated migration of local bacterial EVs to brain, we next administered Aa EVs by applying EV-soaked hydrogels on the gingiva instead of intragingival injection (Fig 3A). It has been demonstrated that an injectable, in situ gelling hydrogel system composed of chitosan and gelatin blends can sustain the release of the drug [23]. The use of hydrogel for the local and sustained delivery of bEVs would simulate the conditions in patients whose periodontal tissues are chronically challenged by bacterial biofilms. When EV-containing hydrogels were applied to sham or ligated mice, PD mice treated with Aa EV-soaked gel exhibited increased Aa EV fluorescence signals measured by IVIS in ventral regions of the brain 12 h after application (Fig 3B–3D). After 24 h of EV-soaked gel administration, the signals were detected in both dorsal and ventral regions of the PD mouse brain in six of the nine mice tested (Fig 3E–3G). Furthermore, EVs appeared more prominently in the PD mice than in the sham control mice after 24 hours of treatment with the Aa EV-soaked gel. (Fig 3H). Consistent with the results obtained with intragingival injection, Aa EV-soaked gel treatment in PD mice induced significantly higher TNF-α and IL-6 expression in both immunofluorescent staining and ELISA compared with non-EV treated and sham control mice (Figs 3I–3K and S3).

To verify successful Aa EV cargo delivery, one of the prominent miRNA-sized small RNA present in Aa EV, Aa-20050 [24], was measured by qRT-PCR. After normalizing the Aa EVs with exogenous Cel-miR-39, which was done due to lack of an internal control for bEV small RNAs, the expression levels of Aa-20050 were found to accumulate in both intragingivally injected and EV-soaked gel-treated mice after 24 h (Figs 2H and 3L).

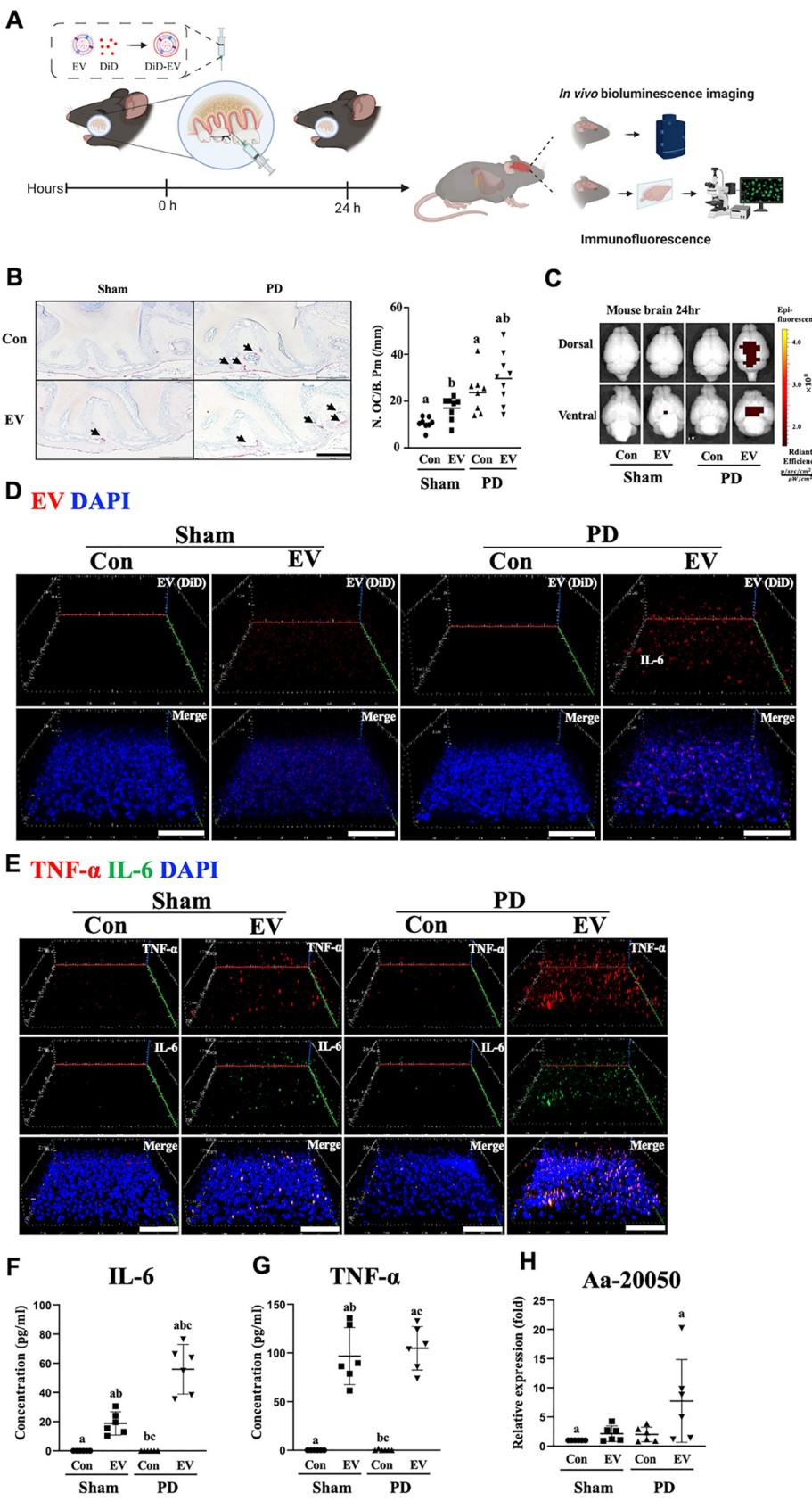

**Fig 2. Aa EVs induced proinflammatory cytokines secretion in periodontal disease (PD) model generated via ligature and subjected to intragingival injection.** (A) Schematic diagram of the in vivo experimental design. DiD-labeled Aa EVs (approximately $2.25 \times 10^9$ particles in 50 µl of PBS) were intragingivally injected (n = 6 per each group). Created with BioRender.com. (B) After 24 h, periodontal tissues were stained for TRAP activity, the representative images of TRAP-positive cells being shown (left panel, scale bar: 200 µm). The number of osteoclasts per bone perimeter (N. OC/B. Pm) and eroded surface per bone surface (ES/BS) were calculated by histomorphometry analysis (right panel). Data are mean ± SD of seven samples. (C) Brain tissues were isolated to measure the accumulated fluorescence signal of DiD-labeled Aa EVs using an in vivo imaging system (IVIS) in the brain. (D) 3D rendering of confocal fluorescence image analysis revealed increased Aa EV particles (red) in the brains (cerebral cortex) of mice with ligature-induced PD. DAPI signals are shown in blue. Scale bar = 80 µm. (E) Confocal image analyses for immunostaining of TNF-α (red), IL-6 (green), and DAPI (blue) in the brains (cerebral cortex) are shown. Scale bar = 80 µm. Also see the S2 Fig. IL-6 (F) and TNF-α (G) quantification using ELISA showed their increased level in response to Aa EV intragingival injection. (H) An Aa-specific small RNA in the brain, Aa-20050, was measured using qRT-PCR following intragingival injection of Aa EVs after 24 h. For qRT-PCR, Cel-miR-39-3p was added (spike-in) for normalization. (F-H) Data are presented as mean ± SD from six independent experiments. One-way ANOVA with Tukey's post-hoc test was used to compare each test group.

## Aa EVs activate the TLR4 and TLR8 signaling pathways

Pathogen-associated molecular patterns (PAMPs) are recognized by Toll-like receptors (TLRs), which are the primary pattern recognition receptors (PRRs) [25]. We previously suggested that Aa EVs can activate the expression of TLR8 via RNA cargo [21]. Another study also showed TLR activation in response to EVs of periodontal pathogens in TLRs expressing-HEK-Blue cells [26,27]. As a gram-negative bacterium, the Aa EV membrane consists of lipopolysaccharide (LPS), a well-known PAMP for TLR4 [28]. Furthermore, owing to its size and abundance, bEV small RNA is a strong candidate PAMP that induces PRR activation and innate immunity [29,30]. Although both TLR7 and TLR8 are the PRRs for ssRNA (single-stranded RNA) in the host cell, TLR8 is less known than TLR7, even though both receptors are expressed on intracellular endosomal membranes and recognize RNA molecules [31]. We evaluated TLR4, TLR7, and TLR8 activation in HEK-Blue cells to verify the pathway underlying Aa EV-induced TLR-mediated TNF-α and IL-6 activation. Activation of TLR8, but not TLR7, was observed in HEK-Blue cells in a dose-dependent manner (Fig 4A–4C). We also compared TLR4, TLR7 and TLR8 activation in DNase-treated Aa EV lysates (DNA-removed EV lysates) and RNase-treated EV lysates (RNA-removed EV lysates) in each TLR-expressing HEK-Blue cells. TLR4 and TLR7 activities in HEK-Blue cells were not affected by either DNase or RNase treatment to the Aa EVs. However, TLR8 activity was considerably down-regulated after RNA removal in Aa EV lysates (Fig 4D–4F). This observation indicated that RNAs within Aa EVs activated the TLR8 signaling pathway rather than the TLR7 pathway.

Next, signaling pathways downstream of TLRs were also investigated. Viral and bacterial RNAs are known to stimulate TLR-MyD88 (myeloid differentiation primary response 88) signaling, leading to the NF-κB activation [32,33], while MyD88 mediates the intracellular signaling of all TLRs with the exception of TLR3 [34]. However, more in-depth research on *MyD88*-deficient cells has shown that there are both MyD88-dependent and -independent pathways that modulate signaling in response to microbial components [35]. Therefore, it is essential to determine whether *MyD88* knockout (KO, -/-) mice have a reduced response to bEV and bacterial exRNAs to confirm the distinct activation pathways for proinflammatory cytokines generated by Aa EVs. Consistently, immunofluorescence staining (Fig 5A) and ELISA (Fig 5B and 5C) assays demonstrated dramatic reduction in the expression of TNF-α and IL-6 after EV challenge and PD induction in the brain of *MyD88* KO mice compared with wild-type (WT). To verify the delivery of Aa EV cargo, Aa-20050 was measured by qRT-PCR as described above (Fig 5D). In addition, NF-κB activity as measured by phospho-p65 and total p65 was not significantly induced in the MyD88 KO group compared to WT, whereas both phospho-

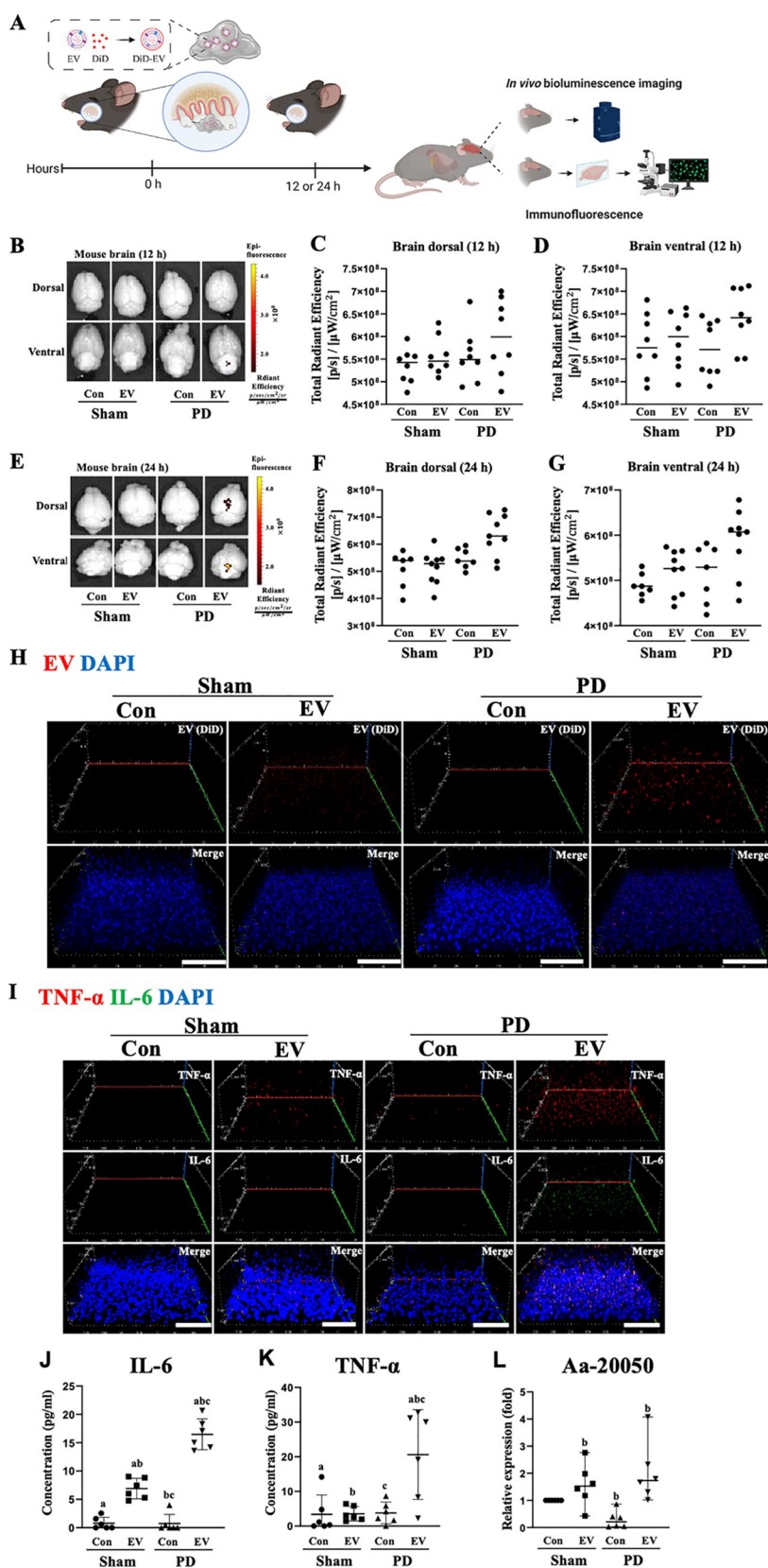

**Fig 3. Aa EV-soaked gel induced proinflammatory cytokine release in a mouse model of ligature-induced periodontal disease (PD).** (A) Schematic diagram of the in vivo experimental design. Gelatin gels (50 μl) including DiD-labeled Aa EVs (approximately $2.25 \times 10^9$ particles in 50 μl of PBS) were administered to mice. Created with BioRender.com. (B-G) Aa EV signals were captured and quantified using IVIS in both dorsal and ventral regions of the brain at 12 h (n = 8 per each group; B-D) and 24 h (n = 7 to 9 per each group; F-G) of EV-soaked gel treatments. (H) 3D rendering of confocal fluorescence image analysis showed increased Aa EV particles (red) in the brains (cerebral cortex) of mice with ligature-induced PD. DAPI signals are shown in blue. Scale bar = 80 μm. (I) Immunostaining of IL-6 (green), TNF-α (red), and DAPI (blue) in the brains (cerebral cortex) is shown. Scale bar = 80 μm. Also see the S3 Fig. IL-6 (J) and TNF-α (K) quantification using ELISA showed their increased level in response to Aa EV intragingival injection. (L) Aa-20050 in the brain was measured using qRT-PCR following intragingival injection of Aa EVs. For qRT-PCR, Cel-miR-39-3p was added (spike-in) for normalization. (J-L) Data are presented as mean ± SD from six independent experiments. One-way ANOVA with Tukey's post-hoc test was used to compare each test group.

p65 and total p65 were activated in the PD mouse group in response to intragingival injection of Aa EVs. (Fig 5E). This observation suggested that Aa EVs can induce proinflammatory cytokine production in the brain via TLR4 and TLR8, mediated by MyD88 with concomitant activation of NF-κB signaling following the onset of periodontitis.

## Aa EVs migrate to trigeminal ganglion via retrograde axonal transport through the maxillofacial nerves

Some viruses, such as herpes simplex virus (HSV), are well characterized to migrate by axonal transmission from the site of infection to local sensory ganglia [36]. Additionally, 16S rDNA of another periodontal pathogen, *Treponema denticola*, was found in the trigeminal ganglia of mice after oral administration of the bacterium [37]. Thus, we examined whether Aa EVs can be directly transported to the neuronal soma through the axon terminal. After epidermal or infraorbital nerve (ION) injection of pre-labeled Aa EVs, the fluorescence signals of Aa EV and RNA cargo were observed exclusively in the V2 region of TG neurons after 24 h (Figs 6A, S4, and S5). However, when the ION of the mice was excised (nerve block), no Aa EV signals were detected following the epidermal injection of EV. Furthermore, while the expression of TNF-α and IL-6 in TG neurons were significantly upregulated by Aa EVs at 24 h in the epidermal-injected group, only very weak or no signals were detected in PBS control and nerve-blocked control groups (Fig 6B and 6C). Because TNF-α and IL-6 expression are reported in glial as well as in neuronal cells [38,39], we used the astrocyte marker glial fibrillary acidic protein (GFAP) and neuronal cell marker (NeuN), along with TNF-α. The immunofluorescence analyses showed that TNF-α was co-expressed with NeuN, but not GFAP in TG, in response to epidermal and intra-ION injection of Aa EVs (Figs 6D and S5). We also discovered that 24 h after the intragingival injection of pre-labeled Aa EVs, the fluorescence signals of Aa EVs and RNA cargo were visible in the TG neurons. Notably, the ligature-induced PD mice showed increased expression of TNF-α and IL-6. In addition, whereas TNF-α and IL-6 expression in TG neurons was significantly increased by Aa EVs at 24 h in PD mice compared to the sham group, dramatically lowered signals were found in both the *MyD88*⁻/⁻ control and PD groups in response to Aa EVs, compared with the significant increase of the cytokines by Aa EV after PD induction in WT mice (S6 Fig). The same observation was made when we applied Aa EVs through an EV-soaked gel (S7 Fig).

## Effect of Aa EVs on the excitability of nociceptive sensory neurons

An increase in proinflammatory cytokines in TG neurons prompted us to examine whether the excitability of nociceptive neurons might be affected by peripheral application of Aa EVs because TNF-α is known to increase the excitability of nociceptive sensory neurons [40–42]. Therefore, we directly examined the excitability of nociceptive sensory neurons isolated from

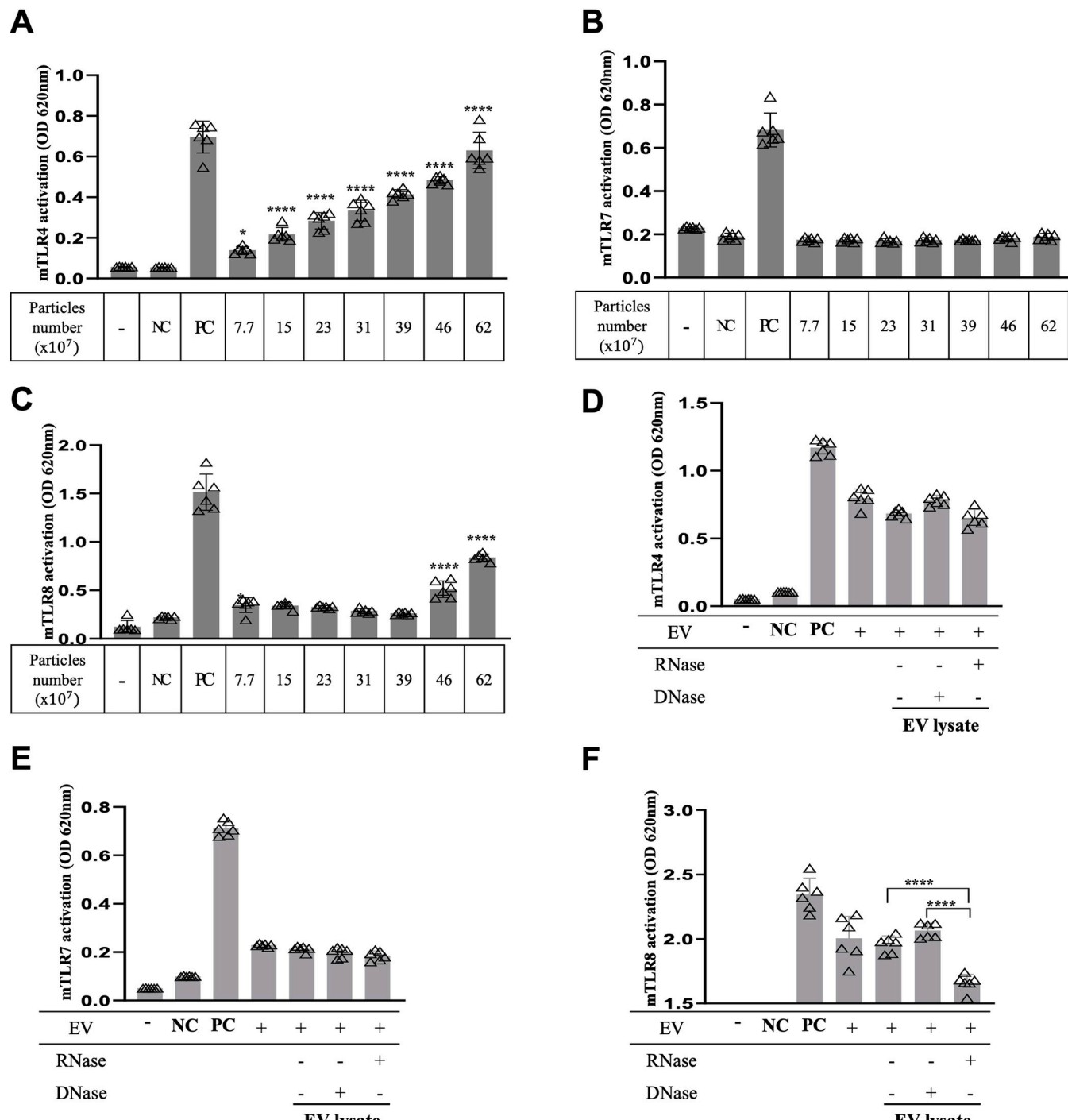

**Fig 4. Aa EVs and RNA cargo activate TLR signaling.** To examine TLR signaling induction by Aa EVs, HEK-Blue reporter cell lines expressing TLR4 (A), TLR7 (B), and TLR8 (C) receptors were used. Cells expressing TLR4 and TLR8, but not TLR7, showed activated signals with increasing Aa EV particle numbers. At a fixed Aa EV particle number ($62 \times 10^7$), TLR4 (D), TLR7 (E) and TLR8 (F) expression was also compared after treatment of RNase or DNase. Expression of TLR4 and TLR7 were unchanged compared to the intact Aa EV particle-treated group. However, TLR8 expression was significantly decreased by RNase-treated Aa EV lysates, which had been treated with the same amount of protein levels as intact EVs (D-F). LPS (100 ng/ml) for TLR4, and R848 (1 μg/ml) for TLR7/8 were used as positive controls (PC) for each cell line. HEK-Blue Null cells were used as a negative control (NC). The data are presented as the mean ± SD from six independent experiments. Significance was calculated by comparison to the non-stimulated control (mock) of the same cell line. n = 6 per each group. *; $p < 0.05$, ****; $p < 0.0001$ (one-way ANOVA with Tukey's post-hoc multiple comparisons test).

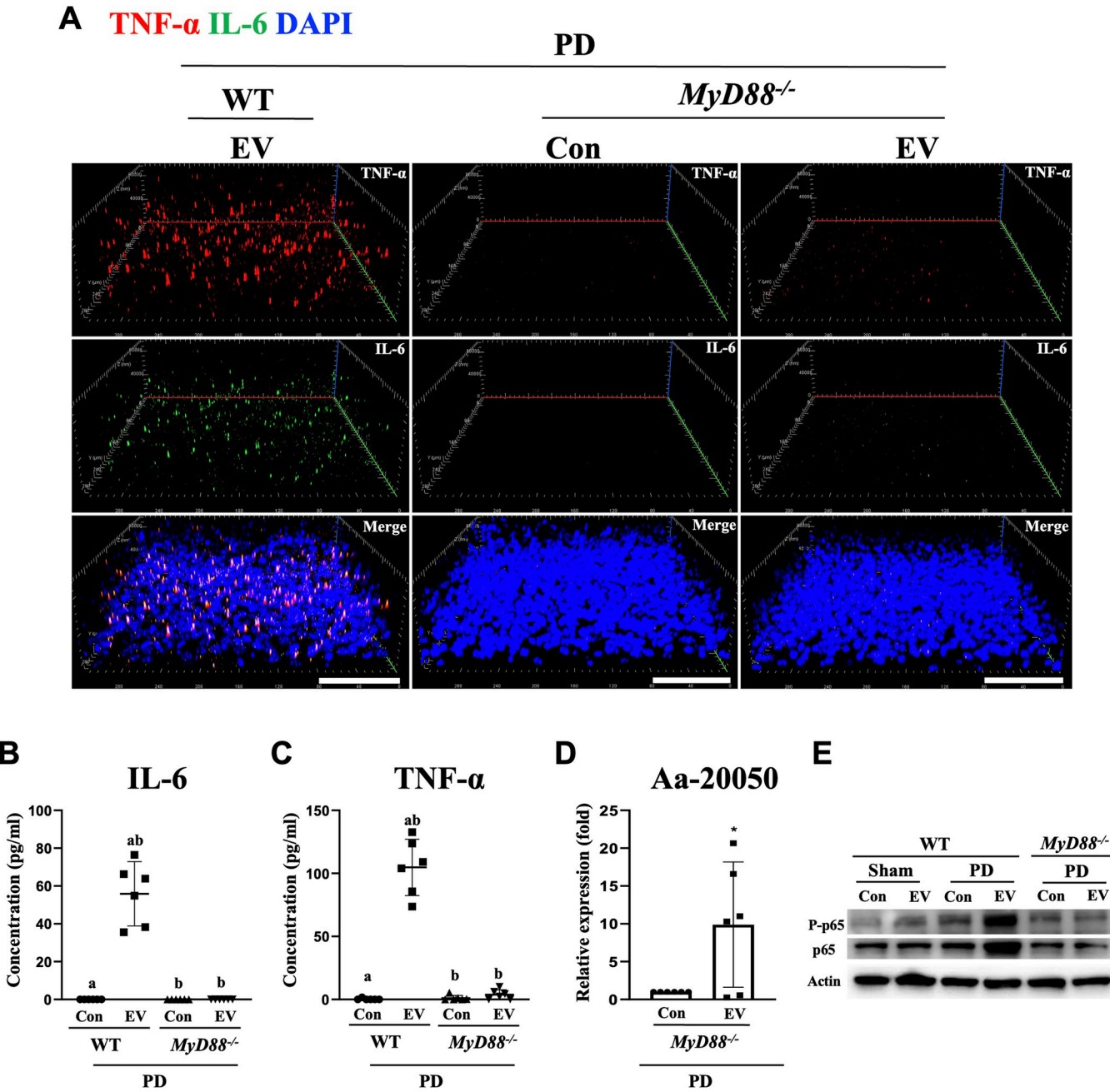

**Fig 5. Aa EV-induced proinflammatory cytokine expression is blocked in *MyD88*[-/-] mice.** DiD-labeled Aa EVs (approximately $2.25 \times 10^9$ particles in 50 μl of PBS) were gingivally injected in WT and *MyD88*[-/-] mice after ligature-induced necrosis (n = 6 per each group). (A) 3D rendering of confocal fluorescence image analysis revealed dramatic reduction of proinflammatory cytokines TNF-α (red) and IL-6 (green) by immunostaining (cerebral cortex area) and ELISA (B and C) in *MyD88*[-/-] after 24 h of injection (A section: Scale bar = 80 μm). (D) Aa-20050 in the brain was measured using qRT-PCR following intragingival injection of Aa EVs. For qRT-PCR, Cel-miR-39-3p was added (spike-in) for normalization. Data are presented as mean ± SD from six independent experiments. (E) Brain tissues were analyzed by western blotting in response to Aa EVs. NF-κB activation (phospho-p65, upper panel) was significantly decreased in the *MyD88*[-/-] mice compared with WT littermate control group at 24 h. Total NF-κB p65 (middle panel) and actin (bottom panel) levels were assessed for loading controls.

Aa EV-treated mice, which were identified by fluorescence signals of Aa EV (red) and RNA cargo (green) (Fig 7A) and the existence of tetrodotoxin-resistant (TTX-R) Na$^+$ currents (Fig 7B). Basal membrane properties of small-sized TG neurons in the PBS-treated and Aa

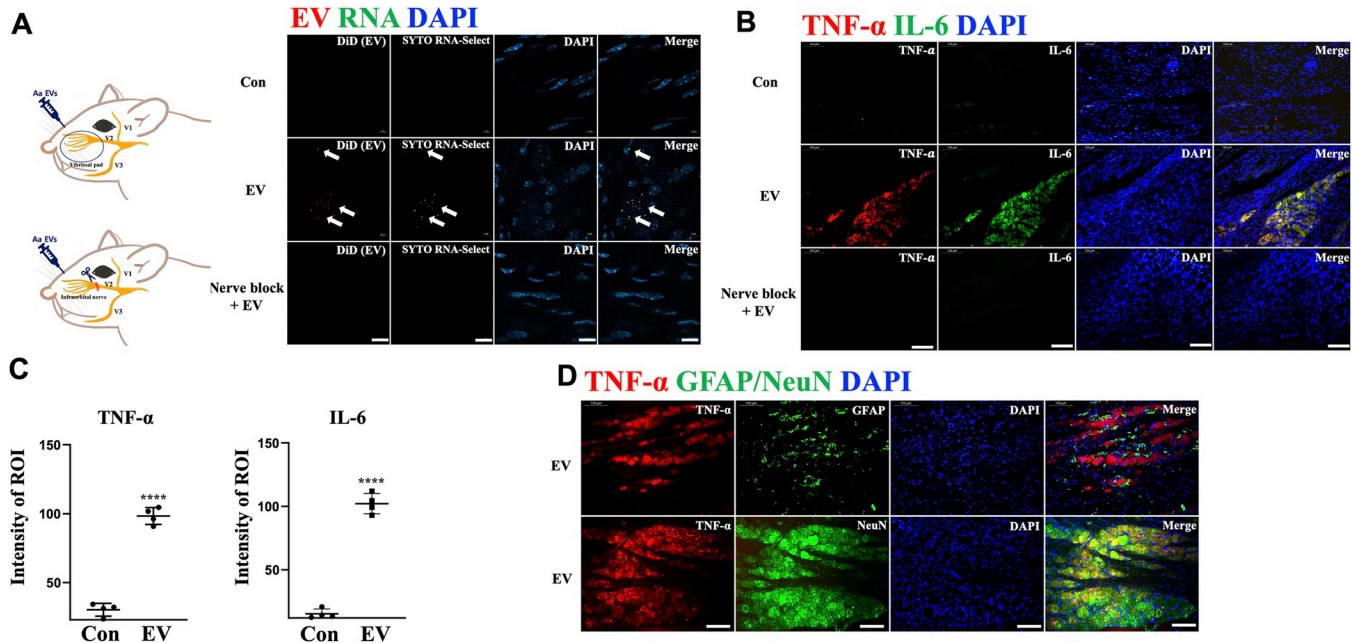

**Fig 6. Aa EVs are directly transmitted by trigeminal ganglion neurons via axon terminal and stimulate the neurons.** (A) Pre-stained Aa EVs (approximately $2.25 \times 10^9$ particles in 50 μl of PBS) with lipid tracer dye DiD (red) and RNA-specific dye SYTO RNA-Select (green) were injected via epidermal injection. The V2 region of trigeminal ganglion, but not the nerve-blocked TG, shows red spots (EV) and green spots (RNA inside EV) 24 h after injection (Scale bar = 10 μm). (B-C) Expression of proinflammatory cytokines TNF-α and IL-6 in the TG were compared with PBS control and ION-blocked TG. TNF-α and IL-6 positive cells were compared in Aa EVs administrated TG with PBS-control following epidermal injection (Scale bar = 100 μm). (D) TNF-α positive cells did not co-localize with astrocyte marker GFAP (green; upper panel), but with neuronal marker NeuN (green; bottom panel) expressing cells following epidermal Aa EVs injection (Scale bar = 100 μm). The data represents four independent biological experiments and are presented as mean ± standard deviation (n = 4 per each group). ****; $p < 0.0001$ (Student's t-test). The observation was also acquired from ION-direct injected TG (S5 Fig).

EV-treated groups are summarized in S1 Table. While the amplitude and shape of action potentials were not different between the PBS-treated control and Aa EV-treated groups (S1 Table), the number of action potentials elicited by depolarizing current stimuli was significantly higher in the Aa EV-treated than in the PBS-treated control groups (Fig 7C and 7D, left panel). Furthermore, the rheobase currents measured in small-sized TG neurons were significantly lower in the Aa EV-treated than in the PBS-treated control groups (Fig 7D, right panel), suggesting that the excitability of nociceptive neurons is higher in the Aa EV-treated than in the PBS-treated control groups.

## Discussion

Periodontitis has been linked to diverse systemic inflammatory conditions including the neuroinflammation often found in AD, Parkinson's disease, and multiple sclerosis [2]. The hematogenous spread of periodontal bacteria and their products due to the compromised physical barriers present in periodontitis or the leakage of inflammatory mediators from periodontal tissues are thought to cause periodontitis-associated systemic inflammation [43]. Although it is controversial whether pathogenic bacteria themselves are able to pass through the BBB and directly induce neuroinflammation, the ability of a pathogen's bEVs to traverse the BBB was demonstrated in our earlier discoveries using a mouse model [20,21]. These findings led us to postulate that bEVs may mediate at least some of the mechanisms through which periodontitis initiates and sustains deleterious brain inflammation. In the present study, we utilized a ligature-induced PD model in mouse that mimic human conditions [44], to test the hypothesis

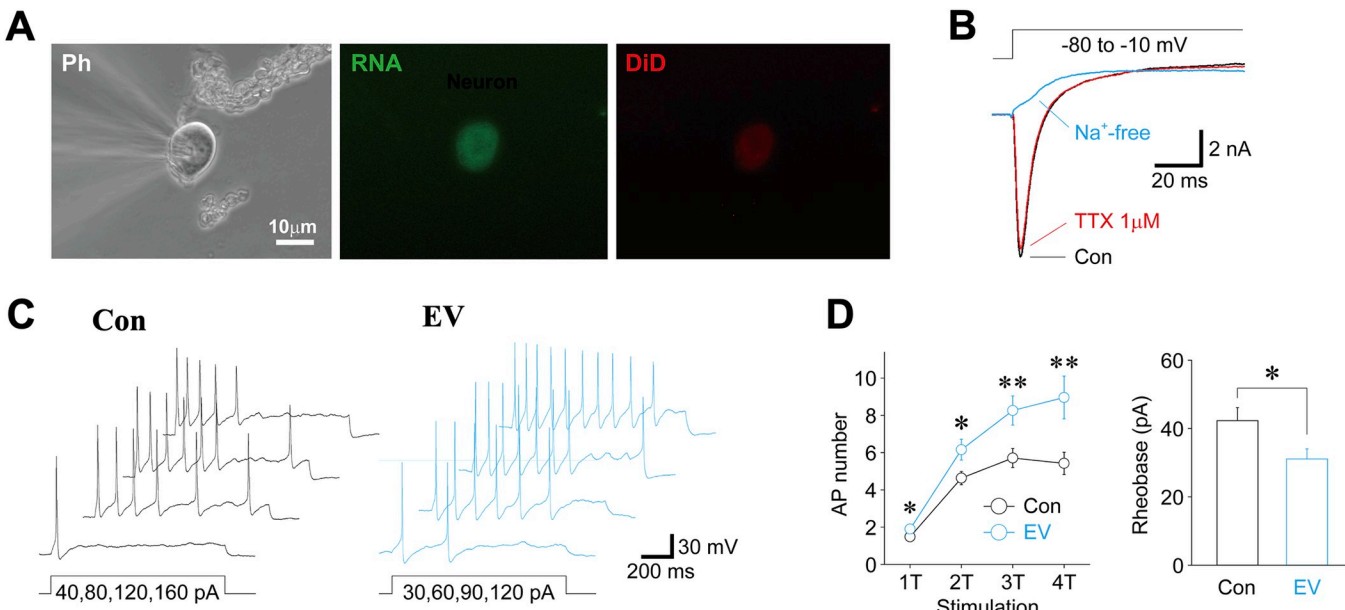

**Fig 7. Peripheral treatment of Aa EVs increases the excitability of nociceptive sensory neurons.** (A) Typical phase contrast (Ph; left) and fluorescence (RNA; middle, DiD; right) images of small-sized TG neurons isolated from Aa EV-injected (approximately $2.25 \times 10^9$ particles in 50 µl of PBS, epidermal injection) mice. (B) Typical raw traces of voltage-gated $Na^+$ current observed from Aa EV-treated small-sized TG neurons in the absence (black) and presence (red) of 1 µM tetrodotoxin (TTX). Voltage-gated $Na^+$ current was elicited by brief voltage step pulses (50 ms duration, -80 mV to -10 mV) in a voltage-clamp condition. Note that the $Na^+$ current recorded in the presence of TTX completely disappeared by adding $Na^+$-free extracellular solution. (C) Typical voltage traces in response to depolarizing current. Four successive raw traces elicited by 1-fold threshold (1T; 40 pA for PBS-treated neuron and 30 pA for Aa EV-treated neuron) to 4-fold threshold (4T) depolarizing current injections were shown. (D) Left, the number of action potentials elicited by depolarizing current stimuli (1T to 4T) in PBS-treated (black) and Aa EV-treated (cyan) small-sized TG neurons. Points and error bars represent the mean and SEM from 52 PBS-treated and 50 Aa EV-treated small-sized TG neurons. *; $p < 0.05$, **; $p < 0.01$ (unpaired t-test). Right, the amplitude of rheobase currents in control PBS-treated (Con, n = 52) and Aa EV-treated (EV, n = 50) neurons. Each column represents the mean and SEM. *; $p < 0.05$, unpaired t-test.

that periodontitis facilitates the migration of oral bEVs to brain to induce neuroinflammation. Indeed, the induction of PD combined with intragingival injection of EV or application of EV-soaked gel significantly increased the transport of Aa EVs to the brain and the induction of TNF-α and IL-6 expression (Figs 2 and 3).

EVs originated from bacteria contain PAMPs such as proteins, lipids, nucleic acids, and polysaccharides that can bind to the host PRRs, including TLRs [45]. We discovered that Aa EVs activated the intracellular TLR8 receptor, in addition to TLR4, which is the cellular membrane receptor for LPS (Fig 4). TLR4 is believed to be the universal PRR for all gram-negative bacterial EVs. Although both TLR7 and TLR8 respond to ssRNA and show high homology to each other at the genetic level, they are supposed to activate separate signaling pathways that result in diverse phenotypes in response to different RNA origins [46]. Furthermore, TLR8 plays an exclusive role in recognizing *Streptococcus Pyogenes* in human monocytes by detecting bacterial RNA [33]. In addition, TLR8 is dynamically expressed during mouse brain development and induces neuronal apoptosis [47]. These studies imply that TLR7 and TLR8 signaling may play distinct roles in the immune system and development. In our experiments, Aa EVs not only selectively stimulated TLR8, but did so in an RNA-dependent manner (Fig 4). If TLR7 and TLR8 have a non-redundant function in recognizing RNAs from distinct sources, then specific RNA modifications or sequence motifs might be essential for selectively activating these TLR receptors. RNA as a constituent of bacterial EVs has gained significant attention because of its multiple regulatory functions [13]. Similar to the case of eukaryotic cells, the majority of the RNAs secreted by bacteria via bEVs (also known as extracellular RNAs or

exRNAs) consist of small RNAs (15–40 nt) [48]. Although several studies have suggested that bEV RNAs may regulate host genes encoding cytokines [24,29] and modulate cellular metabolism [49], it has remained largely unclear whether these small RNAs affect host cells. In this study, the activation of TLR8 by bacterial RNA adds an additional crucial mechanism by which host cells are stimulated by EVs originating from pathogenic bacteria. MyD88 mediates signaling pathways downstream of all TLRs except TLR3 that are primarily responsible for inducing the expression of proinflammatory cytokines [50]. Our data revealed a dramatic reduction in Aa EV-induced TNF-α and IL-6 expression in *MyD88*[-/-] mice. Although our results indicate that TLR-MyD88 signaling is the primary pathway for the expression of proinflammatory cytokines in response to Aa EVs (Fig 5), we cannot rule out the possibility that other TLR-independent endogenous PRR pathways, such as caspase-11, which detects cytoplasmic LPS [51] or other cellular PRRs, may also activate inflammatory signaling pathways in response to bEVs. In neurodegenerative diseases, which are defined by neuronal degeneration and neuronal death in certain parts of the central nervous system (CNS), neuroinflammation by TNF-α and IL-6 is a crucial pathogenic mechanism underpinning neural death [52–54]. In this context, the stimulation of TNF-α and IL-6 production by bacterial EVs via TLR4/8-MyD88 pathways provides a previously unrecognized mechanism that contributes to the development of neuroinflammatory diseases.

The three major nerve branches from the TG, ophthalmic (V1), maxillary (V2), and mandibular (V3), relay painful sensation from the orofacial area. TG is unique among the somatosensory ganglia in terms of its topography, structure, composition, and possibly some functional properties of its cellular components [55]. In addition to glial cells, neuronal cells, particularly TG neurons, also express proinflammatory cytokines like TNF-α under pathological conditions [56–58]. Therefore, we hypothesized that sensory neurons stretched to the maxillary region may take up periodontopathogen EVs and activate proinflammatory signals. Here, we demonstrate for the first time that bEVs can migrate in a retrograde fashion to the soma of TG neurons. Interestingly, neurons, but not the surrounding GFAP-positive astrocytes, showed distinct TNF-α or IL-6 colocalization signals (Figs 6 and S5). In line with these findings, we found that the excitability of nociceptive neurons is higher in the Aa EV-treated than in the control groups (Fig 7). The Aa EV-mediated increase in the excitability of nociceptive neurons could be attributed to Aa EVs regulating ion channels, which could be one of the mechanisms underlying the pain symptoms seen in aggressive and acute periodontitis [59,60]. However, more research is needed to determine how the increased intracellular TNF-α caused by peripheral Aa EVs treatment affects the excitability of nociceptive neurons.

It is unclear how bEVs travel over long distances from nerve terminals to neuronal cell bodies in TG to affect expressional changes required for neuronal activation. One possible mechanism might be the retrograde signaling by "signaling endosomes," a well-known transmission process for some neurotransmitters [61]. Aa EVs transported from the axon terminal of the maxillary region likely remained within the cell body of the V2 TG neuron and were unable to propagate further into the CNS, as we were unable to detect the EVs in any other region of TG following epidermal or intra-ION injection (Figs 6, S4, and S5). How bEVs are transported through the axon and degraded in the neuron requires in-depth investigation.

It has been proposed that the LPS of another periodontopathogen, *Porphyromonas gingivalis*, directly activates trigeminal sensory neurons via TLR4-NF-κB signaling, which may contribute to pulpal pain [62]. Moreover, TLR4 signaling in the trigeminal ganglion stimulates pain induced by acute pulpitis [57]. The retrograde transmission of bEVs from the nerve axon terminal to the neuronal cell body in the current study suggests that bEVs may also contribute to nociception by directly triggering neuroinflammation in the trigeminal ganglion. Interestingly, TLR8 stimulation reportedly increases neuroinflammation after ION injury, which in

turn sustains trigeminal neuropathic pain [63]. Therefore, TLR8 may be a part of the neuronal defense mechanism in the battle against PAMP invasion and induced damage.

Recently, Chacko et al. suggested that *Chlamydia pneumoniae*, a gram-negative respiratory pathogen, can quickly infect the CNS through olfactory and trigeminal neurons, which may play a role in the etiology of Alzheimer's disease [64]. With more reports supporting the association between TG infection and AD [65,66], TG might be an alternative route for pathogen-associated neurodegeneration in neuroinflammatory diseases. However, in addition to AD, our data also point to a potential connection with neuropathic pain. Ample evidence indicates that inflammation is an important mediator of neuropathic pain [67], in which proinflammatory cytokines such as TNF-α and IL-6 are extensively linked to pathological pain with persistent nociceptive responses [68,69]. In view of recent reports of significant up-regulation of TNF-α and IL-6 in trigeminal ganglion in rodent models of trigeminal neuralgia [70,71], it would be interesting to study the trigeminal nociception in PD mice challenged with Aa EVs. Notably, a longitudinal population study suggested that chronic periodontitis patients exhibited significantly higher risk of trigeminal neuralgia than comparison subjects [72].

In summary, periodontopathogen-derived EVs may be closely linked to a variety of neurodegenerative diseases because of their ability to enhance neuroinflammation in the presence of periodontitis (Fig 8). In particular, bEVs can reach the brain through the bloodstream and their direct neuronal transport causes neuroinflammation. TLR/MyD88 signaling-driven proinflammatory cytokine secretion is more prominent in the PD model, indicating a substantial link between periodontal and neuroinflammatory diseases. These findings suggest certain inflammatory disorders may be associated with the translocation of bEVs to systemic organs. Therefore, the ability of bEVs to circulate throughout the body may clarify our understanding of autoimmune disorders, for which up to 50% lack a definite etiological cause [73], while it should be noted that the majority of bEVs can be eliminated in the spleen and liver, as shown in S1 Fig.

Human systemic lupus erythematosus, the etiology of which was previously unknown, has been found to be caused by a TLR7 variant with a gain-of-function that enhances viral RNA sensing and thereby activates autoimmunity [74]. The TLR genetic polymorphism may increase susceptibility to neuroinflammation and trigeminal neuralgia induced by bEVs, a possibility that warrants thorough investigation. Future study is also required to determine ways to prevent bEV-related disorders by inhibiting bEV synthesis and bacterial exRNA invasion of host cells.

## Materials and methods

### Ethics statement

All animal experimental protocols were approved by the Committee on the Care and Use of Animals in Research at Kyungpook National University, Korea (approval number KNU 2022–0246).

### Bacterial culture

*A. actinomycetemcomitans* (ATCC 33384) was inoculated on brain heart infusion (BHI; Difco, Sparks, MD, USA) agar plates and incubated in an anaerobic incubator set at 37°C. After 24 h, the colonies were picked and cultured in BHI medium for 48 h. Picked colonies were also checked for ribosomal DNA to confirm absence of any contamination. The anaerobic incubator was supplied with 5% $CO_2$, 5% $H_2$, and 90% $N_2$. Aa was grown anaerobically in BHI until the desired optical density ($OD_{600}$) was approximately 0.7. Supernatants were collected for EV purification.

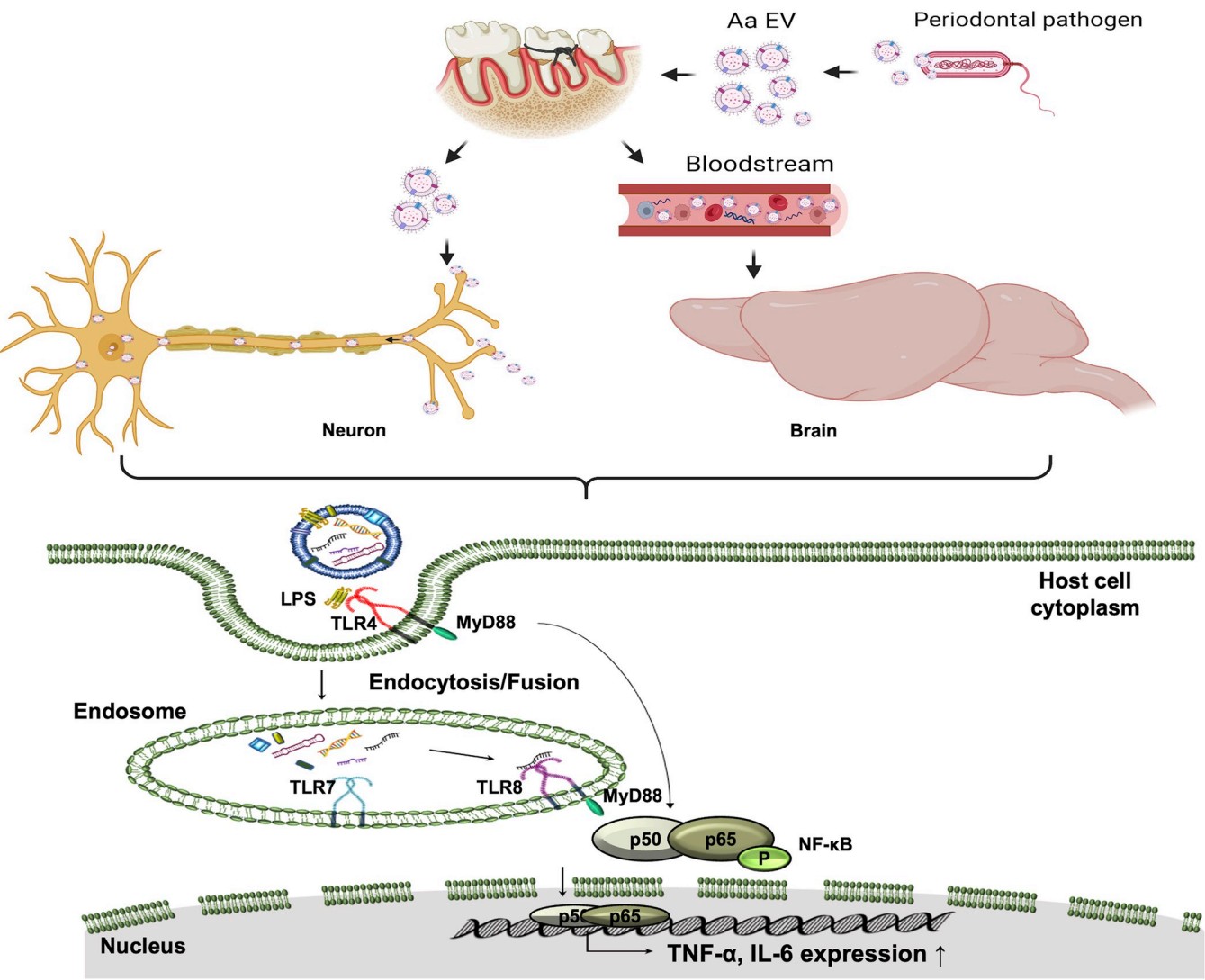

**Fig 8. Graphic summary.** Model of bacterial EV (bEVs) function in the brain and trigeminal neuron. bEVs have the ability to enter host brain cells through the bloodstream and to enter the trigeminal ganglion neuronal cell body through retrograde axonal transport, where they directly trigger proinflammatory neuronal signals. Through the TLR4/TLR8-MyD88 and NF-κB signaling pathways, bEVs and their RNA cargo can stimulate the production of TNF-α and IL-6 in the host cells. The production of neuroinflammation is noticeably more pronounced in the periodontitis disease model, suggesting a strong association between periodontal and neuroinflammatory diseases via bEVs. Created with BioRender.com.

### Isolation and analysis of EVs

The Aa EV samples were prepared and characterized as previously described [20,21]. Briefly, Aa was cultivated anaerobically in BHI until the target optical density was achieved. The bacterial culture supernatant was then separated by centrifugation at 6000 g for 15 min at 4°C. To eliminate any remaining bacteria, the supernatant was filtered using a 0.22-µm pore filter (Sigma-Aldrich, St. Louis, MO, USA). Next, the supernatant was concentrated through ultrafiltration employing a MasterFlex L/S complete pump system (Cole-Parmer, Vernon Hills, IL, USA) and a 100 kDa Pellicon 2 mini Ultrafiltration module (Sigma-Aldrich). Finally, the OMV was extracted from the filtered and concentrated supernatant using ExoBacteria OMV Isolation Kits (SBI, Mountain View, CA, USA), following the manufacturer's instructions. For staining, the EVs (resuspended in 100 µl phosphate buffered saline [PBS], without calcium and

magnesium; GE Healthcare BioSciences, Pittsburgh, PA, USA) were incubated with 10 μM red fluorescent Lipophilic Tracer DiD (1,1'-dioctadecyl-3, 3, 39, 39-tetramethylindodicarbocyanine, 4-chlorobenzenesulfonate salt; Thermo Fisher Scientific, Waltham, MA, USA) and/or 2 mM SYTO RNA-Select Green Fluorescent Cell Stain Kit (Thermo Fisher Scientific) for 1 h at 37˚C to stain RNAs inside EV. To exclude residual dyes, the samples were washed once with 1 ml of PBS followed by ultracentrifugation at $150,000 \times g$ for 1 h at 4˚C. No pellets were visible when the same protocol was applied to the DiD dye solution without Aa EVs.

For the physical Aa EV lysis, 1 ml of the Aa EV was frozen and thawed 5 times followed by sonication for 30 s and then placed in ice for 30 s. Subsequently, 1 μl of RNase A or DNase I was added to 1 ml of the lysed EV and incubated at 37˚C for 30 min. The purified EVs were stored at -80˚C until needed.

The EVs were analyzed via nanoparticle tracking analysis using the NanoSight system (NanoSight NS300; Malvern Panalytical, Malvern, United Kingdom) according to the manufacturer's protocols as described previously [21]. The samples were diluted 100-fold with PBS to a total volume of 1 ml. Particle size was measured based on Brownian movement, and the particle concentration was quantified in each sample. EVs were quantified using Bradford reagent (Bio-Rad, Hercules, CA, USA) and the NanoSight system.

For transmission electron microscopy, the purified EV samples were diluted 20-fold in PBS and applied to 200-mesh Formvar/Carbon grids (Ted Pella, Redding, CA. USA) without negative staining. They were then viewed with an electron microscope (HT7700; Hitachi, Tokyo, Japan) operated at 100 kV.

## Animals and experimental groups

Male C57BL/6J mice (Central Laboratory Animal, Seoul, Korea) were housed in a specific pathogen-free facility with regulated temperature/humidity and 12-hour light/dark cycle. The animals had free access to normal rodent chow and tap water. The *MyD88*-deficient mice (B6.129P2-*Myd88*<sup>tm1.1Defr</sup>/J) were obtained from the Jackson laboratory (Bar Harbor, ME, USA). and backcrossed into C57BL/6J background for more than six generations before use. Littermate WT mice were used as controls. The study was performed in accordance with the ARRIVE (animal research: reporting of in vivo experiments) guidelines [75]. The ligature-induced PD model was generated as previously described [76]. Briefly, silk ligatures (6–0 non-absorbable braided silk, Ailee, Busan, Korea) were placed around the second molar of the left maxilla of male C57BL/6 mice (8-week-old) under anesthesia. After ligation, a total of $2.25 \times 10^9$ particles ($4.5 \times 10^{10}$ particles/ml) Aa EVs in 50 μl of PBS were either injected into the gingiva of the left maxilla or applied to the ligated area in the periodontium in a hydrogel manufactured as described below. The induction of periodontitis was verified by the formation of TRAP-positive osteoclasts in alveolar bones as described previously [76]. The tissue samples were decalcified and embedded in paraffin before the preparation of coronal sections of 5-μm thickness around the ligation site using a microtome (RM2245, Leica Microsystems, Wetzlar, Germany). Tissue sections were then subjected to TRAP staining using an acid phosphatase leukocyte kit (Sigma-Aldrich) for the visualization of TRAP-positive cells. For quantitative assessments, histomorphometric analyses were conducted in a masked manner to determine the number of osteoclasts per bone perimeter (N. OC/B. Pm) and the eroded surface per bone surface (ES/BS). These measurements were made using the OsteoMeasure software system (OsteoMetrics, Decatur, GA, USA).

EV-soaked hydrogel was generated as previously described [23]. Briefly, chitosan (448869-50G, Sigma-Aldrich) solution was obtained by dissolving 200 mg chitosan in 0.1 M HCl (10 ml). Gelatin (G9391-100G, Sigma-Aldrich) was added to deionized water and heated at 50˚C

for 2 h to obtain 3% w/v gelatin. The gelatin solution was dropped into the chitosan solution in an ice bath with magnetic stirring and stirred for 15 min to obtain a homogeneous mixture. Then, the cooled 1 ml NaHCO$_3$ (1 M) was dropped into the stirred chitosan/gelatin solution in an ice bath to adjust the pH between 7–8. The resulting solution was stirred for 30 min to obtain a homogeneous mixture. The hydrogel was formed by heating the chitosan/gelatin solution in a water bath at 37°C for 5 min. Next, an EV-soaked gel was prepared using Aa EVs, and 50 μl of this gel, which contained an equivalent number of EV particles ($2.25 \times 10^9$) to those used for injection, was applied either to the ligated area in the model mice or to the same region in the control mice.

Aa EVs were introduced by epidermal or ION injection to determine whether bEVs can be directly transferred to the brain via axon terminals of neurons. The mice were deeply anesthetized with a cocktail mixture of ketamine (80 mg/kg) and xylazine (10 mg/kg). After shaving and local sterilization of the skin, Aa EVs or PBS were injected into the left cheek (vibrissal pad) or ION using a Hamilton syringe and syringe pump (LEGATO 130 syringe pump, infusion rate, 2 μl/min; KD Scientific Inc., MA, USA). ION surgery was performed using the FRICT-ION method, as described by Montera and Westlund [77]. The incision was small enough to not require wound closure, and tissue adhesive (3M Vetbond; 3M, Minnesota, USA) was used to facilitate complete closure. The mice were sacrificed 24 h later and infused for further immunohistochemical analysis.

For electrophysiology, animals (C57BL/6J mice, 7–8 weeks old, male) were anesthetized with a cocktail mixture of ketamine (80 mg/kg) and xylazine (10 mg/kg). After shaving and local sterilization of the skin, the Aa EVs or PBS were injected into the left cheek (vibrissal pad; epidermal injection) using the Hamilton syringe. After 24 h injection of the Aa EVs or PBS, the mice were decapitated under ketamine anesthesia (100 mg/kg, intraperitoneal injection). The maxillary division of TG was dissected and treated with a standard external solution (in mM; 150 NaCl, 3 KCl, 2 CaCl$_2$, 1 MgCl$_2$, 10 glucose, and 10 Hepes, [pH 7.4 with Tris-base]) containing 0.3% collagenase (type I) and 0.3% trypsin (type I) at 37°C for 40–50 min. Thereafter, the TG neurons were dissociated mechanically by trituration with fire-polished Pasteur pipettes in a culture dish (CELLSTAR TC, Greiner Bio-One, Chonburi, Thailand). The isolated TG neurons were used for electrophysiological recordings 1–6 h after preparation.

## In vivo tracking of Aa EVs

To confirm the presence of Aa EVs, the perfusion organs were imaged in vitro. Stained Aa EVs were introduced into C57BL/6J mice via tail vein injections. After 12 or 24 h of injection in the living mice, major organs including the heart, liver, spleen, kidneys, lungs, and brain tissue were imaged. The fluorescence intensity in organ tissues was determined semi-quantitatively using a Caliper IVIS (In Vivo Imaging System) Lumina in vivo imager (Caliper Life Sciences, Boston, MA, USA). Bioluminescence was quantified using Living Image v.4.2 software (Perkin Elmer, Beaconsfield, UK) and corrected for background bioluminescence.

## Immunofluorescence staining

For tissue staining, including whole brain, tissues were initially cleared using a high-resolution tissue-clearing kit (HRTC-001; Binaree, Daegu, South Korea). Briefly, the uncleared tissues were fixed for more than 12 h in 4% paraformaldehyde in PBS. Next, the fixed tissues were immersed in a fixing solution for 24 h and then incubated with tissue-clearing solution in a shaking incubator at 35°C for 48 h. After three rinses in the washing solution, the washed tissues were incubated in mounting and storage solution (SHMS-060; Binaree). Tissues were then cut at 100 μm thickness with a vibratome (VT1200S; Leica Biosystems, Wetzlar,

Germany). For TG immunostaining, experimental mice were perfused transcardially with 4% paraformaldehyde for 20 min. Isolated tissues were then immersed with 4% paraformaldehyde for 1 hour 40 min and incubated in cryoprotective solution (30% sucrose in PBS) overnight at 4˚C. Cryotome sections were cut at 20 μm thickness with a cryostat (SM2010R; Leica Biosystems, Wetzlar, Germany) and penetrated with 50% ethyl alcohol for 30 min, blocked, and incubated with the following respective antibodies. For primary immunostaining, anti–TNF-α (1:200, ab6671; Abcam), anti–IL-6 (1:200, AF406NA; R&D Systems), anti-GFAP (1:400, #3670s, Cell Signaling Technology, Danvers, MA, USA), and anti-NeuN (1:100; MAB377, Merck, Billerica, MA, USA) antibodies were applied overnight. The tissues were then washed extensively in PBS (3 times for 10 min) and incubated with the following specific secondary antibodies to respective primary antibodies: goat anti-rabbit IgG (Alexa Fluor 594, ab150080; Abcam), donkey anti-goat IgG (Alexa Fluor 488, ab150077; Abcam), Alexa Fluor 594 AffiniPure F(ab')₂ Fragment Goat Anti-Rabbit IgG (H+L) (1:200, 111-586-144, Jackson ImmunoResearch, West Groove, PA), Alexa Fluor 488 AffiniPure F(ab')₂ Fragment Donkey Anti-Goat IgG (H+L) (1:200, 705-546-147, Jackson ImmunoResearch, West Groove, PA, USA), Donkey anti-Goat IgG (H+L) Cross-Adsorbed Secondary Antibody, Alexa Fluor 555 (1:1,000, A-21432, Invitrogen, Waltham, MA, USA), Goat anti-Mouse IgG (H+L) Cross-Adsorbed Secondary Antibody, and Alexa Fluor 488 (1:2,000, A-11001, Invitrogen) were used for fluorescence detection. The tissue sections were washed and mounted on microscope slides with VECTASHIELD Antifade Mounting Medium with 4′,6′-Diamidino-2-phenylindole (DAPI; H-1200, Vector Laboratories, Newark, CA, USA) to preserve fluorescence and nuclear staining. Fluorescence was imaged using a digital camera (DFC425C, Leica Biosystems, Wetzlar, Germany) attached to a Leica DM IL (Leica Biosystems) or a laser scanning confocal microscope (LSM Zeiss 800; Carl Zeiss Microscopy, Jena, Germany) equipped with an objective lens. Fluorescence was excited with 488, 594, and 638 nm lasers, and emission was detected with 505, 545, and 660 nm band-pass filters. The scanning laser confocal microscope image was composed of 150 adjacent planes, forming a z-stack. The distance between the planes (z-stack) was set to 0.5 μm. Images were acquired with 10× (pixel size 0.56 μm) and 20× (pixel size 0.28 μm) panoramic lenses using the tail-scan mode with a three-axis motorized stage to cover the entire coronal section of the tissues. All LSFM image data were saved in.czi file type of Zen software (Carl Zeiss), and images were reconstructed and 3-dimensionally–rendered using Arivis Vison 4D software (Arivis AG, Munich, Germany) and Imaris software (Bitplane, Belfast, United Kingdom).

## Enzyme-linked immunosorbent assay (ELISA) of cytokines

Levels of inflammatory cytokines were determined in the mouse tissues. Tissue samples were analyzed using the ELISA kits for IL-6 (KET7009; Abbkine, Wuhan, China) and TNF-α (ADI-900- 047; Enzo, Madison Avenue, New York, NY, USA), according to the manufacturer's instructions. Standard solutions were prepared using the reagents provided in the respective kits. The plates were read at 450 nm using an Emax Plus microplate reader (Molecular Devices, San Jose, CA, USA). Proteins were quantified using mean values of duplicate samples.

## qRT-PCR for Aa-20050 small RNA

We determined the expression of Aa small RNA Aa-20050 using qRT-PCR to confirm the infection of Aa EVs in mouse tissues because it is one of the most highly expressed in Aa EV and can be detected by qRT-PCR [24]. Cel-miR-39 was used as a spike-in control and for normalization [78].

For Aa-20050 qRT-PCR, 3 μl synthetic Cel-miR-39-3p (330 fmol/ml stock; Bioneer, Daejeon, Korea) was added during RNA extraction, and total RNA (25 ng) was reverse-transcribed using a TaqMan MicroRNA Reverse Transcription Kit (Applied Biosystems, Foster City, CA, USA). From the 15 μl reaction mixture, 1.33 μl was used for qRT-PCR, which was performed using the TaqMan Universal PCR Master Mix (Applied Biosystems) and specific TaqMan probes (Applied Biosystems). Each RNA sample was prepared in a reaction volume of 20 μl. PCR was performed in 96-well plates using the 7500 Real-Time PCR System (Applied Biosystems). The expression of each Aa-20050 was determined from three replicates for each RNA sample.

## TLR reporter assay

To evaluate the ability of Aa EVs to stimulate innate immune responses, we used HEK-Blue mTLR4, mTLR7, and mTLR8 cells (InvivoGen, San Diego, CA, USA) harboring an NF-κB-inducible secreted embryonic alkaline phosphatase (SEAP) reporter gene. Up to $5 \times 10^4$ HEK-Blue-mTLR4, mTLR7, and mTLR8 cells suspended in 200 μl HEK-Blue detection medium (InvivoGen) were mixed with Aa EV at $7.7 \times 10^7$–$6.2 \times 10^8$ particles in a 96-well cell culture plate. Untreated cells in HEK-Blue detection medium were used as controls. After 6 and 16 h incubation at 37˚C in 5% $CO_2$, SEAP activity was determined at 655 nm according to the manufacturer's recommendations (InvivoGen). All samples were measured in triplicate. Stimulation of TLR4, TLR7, and TLR8 receptors was expressed relative to the SEAP activity level in untreated control cells.

## Western blot analysis

For NF-κB p65 and phosphorylated (phospho)-p65 western blotting, 20 μg whole-cell lysates were subjected to SDS-PAGE, followed by transfer to a PVDF membrane. The membrane was blocked in 5% skim milk/0.1% Tris-buffered saline–Tween 20 at room temperature for 30 min, followed by incubation with a 1:1000 final diluted NF-κB p65 (8242; Cell Signaling Technology) and phospho-p65 antibodies (Ser536, 3033; Cell Signaling Technology). This was followed by incubation with an anti-rabbit secondary antibody conjugated with horseradish peroxidase (7074; Cell Signaling Technology). β-actin was used as an internal control and its expression was determined using a specific antibody (1:1,000, SC-47778, Santa Cruz Biotechnology Inc., Santa Cruz, CA, USA).

## Electrophysiology

All electrophysiological measurements were performed using a patch-clamp amplifier (Multiclamp 700B; Molecular Devices, Union City, CA, USA). In current-clamp experiments to record membrane voltage, patch pipettes were made from borosilicate capillary glass (1.5 mm outer diameter, 0.9 mm inner diameter; G-1.5; Narishige, Tokyo, Japan) using a pipette puller (P-97; Sutter Instrument Co., Novato, CA, USA). The resistance of the recording pipettes filled with internal solution (145 mM KF, 10 mM KCl, 2 mM EGTA, and 10 mM HEPES; pH 7.2 with Tris-base) was 1.0–1.5 MΩ. The liquid junction potential was corrected. The pipette capacitance and series resistance were compensated for (40%–70%). In the end of all current clamp experiments, TG neurons were held at a holding potential of –80 mV in a voltage-clamp mode, and brief depolarizing step pulses (100 ms duration, up to –10 mV) were applied to verify the existence of TTX-R $Na^+$ channels. Membrane voltage and current were low-pass filtered at 2–5 kHz and acquired at 10–20 kHz (Digidata 1550; Molecular Devices). All electrophysiological experiments were performed at room temperature (22–25˚C). All drugs and reagents were purchased from Sigma-Aldrich. The overshoot, half width, hyperpolarization, and time

constant of hyperpolarization of single action potentials were analyzed using a Clampfit program (version 10.7, Molecular Devices). The rheobase current, which is the minimal threshold current to trigger action potentials at patched neurons, was determined by successive depolarizing current stimuli (5–10 pA increments, 500 ms duration). The number of action potentials elicited by 4-successive depolarizing current stimuli [integers of the threshold currents (1T to 4T), 1 s duration] was counted in control PBS-treated and EV-treated small-sized TG neurons.

## Statistical analysis

Data are presented as mean ± standard deviation (SD) or mean ± standard error of the mean (SEM), where applicable. Differences among sample group values were analyzed using one-way ANOVA followed by Tukey post-hoc test. Significant difference between two groups was calculated by the parametric 2-tailed, unpaired Student's t-test. All analyses were conducted using Prism software 8.4.2 (GraphPad Software, San Diego, CA, USA), and statistical significance was set at $p < 0.05$.

## Supporting information

**S1 Fig. Aa EVs can spread to systemic organs following intravenous tail vein injection.** (A) General scheme of the in vivo experimental design. Purified Aa EVs were quantified using NTS device and calculated particle numbers were used for each injection. Six-week-old male mice were intravenously administered lipid tracer, DiD-labeled Aa EVs (approximately $2.25 \times 10^9$ particles in 50 μl of PBS) for the indicated period. Created with BioRender.com. (B) The brain, heart, lung, kidney, spleen, and liver were resected, and the distributions of DiD-labeled Aa EVs were determined using an in vivo imaging system (IVIS). Although weaker fluorescence intensity was observed in the brain and lungs than in the spleen and liver, the signals became strong after 12 to 24 h of i. v. injections; IVIS signals in the brain were measured separately. Previously, we showed that Aa EVs and exRNAs specifically induce IL-6 expression in microglial cells and TNF-α expression in the brain. These proinflammatory cytokines were measured in the spleen and liver using immunofluorescence staining. Confocal image analysis was performed to detect IL-6 (green) and TNF-α (yellow) in spleen (C) and the liver (G). Red spots indicate Aa EVs. Spleen IL-6 (D), spleen TNF-α (E), liver IL-6 (H), and liver TNF-α (I) were quantified using ELISA. The levels of TNF-α and IL-6 were elevated in the spleen 12 and 24 h after injection, but not in the liver. To verify the successful Aa EV cargo delivery, one of the prominent miRNA-sized small RNAs present in Aa EV, Aa-20050, was measured by qRT-PCR in the spleen (F) and liver (J). For qRT-PCR, Cel-miR-39-3p was added (spike-in) for normalization and found to accumulate in the liver after 24 h of injection but not in the spleen; this may be attributed to rapid degradation of small RNA in these organs due to its high RNase activity. Data are presented as mean ± SD from five independent experiments. Scale bar = 100 μm. Depending on the organ, it may be possible for it to serve as both a target and an elimination organ. Most Aa EVs accumulate in the spleen and liver but induce weaker proinflammatory cytokine release, suggesting that the liver and spleen might act as elimination organs for blood-borne bEVs and their cargo, including RNAs.
(TIFF)

**S2 Fig. Aa EVs-induced proinflammatory cytokines secretion in PD model generated via ligature and subjected to intragingival injection.** DiD-labeled Aa EVs (approximately $2.25 \times 10^9$ particles in 50 μl of PBS) were intragingivally injected to mice for 24 h. Confocal fluorescence image analysis showed increased Aa EV particles (red) in the brains of mice with ligature-induced PD. DAPI signals are shown in blue. Left panel is the sagittal section of the

brain, showing a merged fluorescence image of Aa EV-injected PD mice. Boxes (A, cerebral cortex; B, hippocampus; C, cerebellum) in the middle image are magnified in the right panel.
(TIFF)

**S3 Fig. Aa EV-soaked gel-induced proinflammatory cytokine release in a mouse model of ligature-induced PD.** Gelatin gels including DiD-labeled Aa EVs (approximately $2.25 \times 10^9$ particles in 50 μl of PBS) were administered to mice for 24 h. Confocal fluorescence image analysis showed increased Aa EV particles (red) in the brains of mice with ligature-induced PD. DAPI signals are shown in blue. Left panel is the sagittal section of the brain, showing a merged fluorescence image of Aa EV-injected PD mice. Boxes (A, cerebral cortex; B, hippocampus; C, cerebellum) in the middle image are magnified in the right panel.
(TIFF)

**S4 Fig. Aa EVs are directly transmitted by trigeminal ganglion (TG) neurons via axon terminal and stimulate the neurons.** The TG, which has three major branches: the ophthalmic (V1), maxillary (V2), and mandibular nerves (V3), relays painful sensation from the orofacial area and is unique among the somatosensory ganglia in terms of its topography, structure, composition, and possibly some functional properties of its cellular components. Epidermal injected Aa EVs were directly taken up by TG neurons and those Aa EVs activated TNF-α, specifically in the V2 region of TG. Boxes in the upper panel image are magnified in the lower panel. Scale bar = 100 μm.
(TIFF)

**S5 Fig. Aa EVs are directly transmitted by trigeminal ganglion (TG) neurons via axon terminal and stimulate the neurons.** (A) Pre-stained Aa EVs with lipid tracer dye DiD (red) and RNA-specific dye SYTO RNA-Select (green) were injected via direct infraorbital nerve (ION) injection. The V2 region of trigeminal ganglion, but not the nerve blocked TG, shows red spots (EV) and green spots (RNA inside EV) 24 h after injection (Scale bar = 10 μm). (B-C) Expression of proinflammatory cytokines TNF-α and IL-6 in the TG was compared with PBS control and ION-blocked TG. TNF-α and IL-6 positive cells were compared in Aa EVs administrated TG with PBS-control (Con) following ION injection (Scale bar = 100 μm). (D) TNF-α positive cells did not co-localize with astrocyte marker GFAP (green; upper panel), but did with neuronal marker NeuN (green; bottom panel) expressing cells following epidermal Aa EVs injection (Scale bar = 100 μm). (E) IL-6 (red) positive cells did not co-localize with astrocyte marker GFAP (green; upper panel), but did with neuronal marker NeuN (green; bottom panel) expressing cells following epidermal Aa EVs injection (Scale bar = 100 μm). The data represents four independent biological experiments and are presented as mean ± standard deviation (SD). ***; $p < 0.001$, ****; $p < 0.0001$ (Student's t-test).
(TIFF)

**S6 Fig. Aa EVs are directly transmitted by trigeminal ganglion (TG) neurons by gingival and Aa EV-induced proinflammatory cytokine expression is blocked in TG of $MyD88^{-/-}$ mice through intragingival administration.** DiD-labeled Aa EVs (approximately $2.25 \times 10^9$ particles in 50 μl of PBS) were injected in WT and $MyD88^{-/-}$ mice after ligature-induced necrosis. (A) General scheme of the in vivo experimental design. Pre-stained Aa EVs with lipid tracer dye DiD (red) and RNA-specific dye SYTO RNA-Select (green) were injected via intragingival injection. Created with BioRender.com. (B-C) The V2 region of trigeminal ganglion shows red spots (EV) and green spots (RNA inside EV) at 12 h (B) 24 h (C) after injection (scale bar = 10 μm). (D-E) Fluorescence microscopic image analysis revealed increased TNF-α (red) and IL-6 (green) level of TG in response to Aa EV intragingival injection, with a dramatic reduction of proinflammatory cytokines, TNF-α (red) and IL-6 (green) were observed by

immunostaining in *MyD88*[−/−] after 24 h of injection (Sham Con, n = 3; Sham EV, n = 3; PD Con, n = 3; PD EV, n = 4; *MyD88*[−/−] Con, n = 3; *MyD88*[−/−] EV n = 2). TNF-α and IL-6 positive cells were compared in Aa EV-administrated TG with PBS-control (Con) following intragingival injection (ND, non-detectable). The graphs are presented as mean ± standard deviation (SD). One-way ANOVA with Tukey's post-hoc test was used to compare each test group. (TIFF)

**S7 Fig. Aa EV-soaked gel induced proinflammatory cytokine release in trigeminal ganglion (TG) neurons of a mouse model of ligature-induced periodontal disease (PD).** (A) Schematic diagram of the in vivo experimental design. Gelatin gels including Aa EVs (approximately $2.25 \times 10^9$ particles in 50 μl of PBS) were administered to mice (Sham Con, n = 3; Sham EV, n = 3; PD Con, n = 3; PD EV, n = 3). Created with BioRender.com. (B) Immunostaining of TNF-α (red), IL-6 (green), and DAPI (blue) in the TG is shown. Scale bar = 100 μm. (C) TNF-α and IL-6 in TG quantification using ELISA showed their increased level in response to Aa EV-soaked gel. The graphs are presented as mean ± standard deviation (SD). One-way ANOVA with Tukey's post-hoc test was used to compare each test group. (TIFF)

**S1 Table. Passive and active membrane properties of small-sized trigeminal ganglion (TG) neurons.** (DOCX)

## Acknowledgments

The authors thank Drs. Scott Young (NIH) and Eph Tunkle for critical reading of the manuscript.

## Author Contributions

**Conceptualization:** Youngkyun Lee, Heon-Jin Lee.

**Data curation:** Jae Yeong Ha, Jiwon Seok, Suk-Jeong Kim, Hye-Jin Jung, Ka-Young Ryu.

**Formal analysis:** Il-Sung Jang, Su-Hyung Hong, Youngkyun Lee, Heon-Jin Lee.

**Funding acquisition:** Youngkyun Lee, Heon-Jin Lee.

**Investigation:** Jae Yeong Ha, Jiwon Seok.

**Methodology:** Jae Yeong Ha, Jiwon Seok, Suk-Jeong Kim, Hye-Jin Jung, Ka-Young Ryu, Michiko Nakamura, Il-Sung Jang.

**Project administration:** Youngkyun Lee, Heon-Jin Lee.

**Resources:** Il-Sung Jang, Su-Hyung Hong, Youngkyun Lee, Heon-Jin Lee.

**Supervision:** Heon-Jin Lee.

**Validation:** Il-Sung Jang, Su-Hyung Hong, Youngkyun Lee, Heon-Jin Lee.

**Visualization:** Jae Yeong Ha.

**Writing – original draft:** Heon-Jin Lee.

**Writing – review & editing:** Jae Yeong Ha, Jiwon Seok, Suk-Jeong Kim, Hye-Jin Jung, Ka-Young Ryu, Michiko Nakamura, Il-Sung Jang, Su-Hyung Hong, Youngkyun Lee, Heon-Jin Lee.

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
