## [Decision Letter · Decision Letter 0]

24 Jul 2023

Dear Prof. Lee,

Thank you very much for submitting your manuscript "Periodontitis promotes bacterial extracellular vesicle-induced neuroinflammation in the brain and trigeminal ganglion" for consideration at PLOS Pathogens. As with all papers reviewed by the journal, your manuscript was reviewed by members of the editorial board and by several independent reviewers. In light of the reviews (below this email), we would like to invite the resubmission of a significantly-revised version that takes into account the reviewers' comments.

Thank you for submitting your work to PLOS Pathogens and I am sorry for the delay getting the reviews back to you. The Reviewers found the work of interest but there are some major issues that need to be addressed with additional experimental evidence. In particular, I agree with Reviewer 1 that specificity of the response needs to be addressed. Do other EVs have the same response as Aa EVs and are they trafficked similarly? These controls need to be demonstrated. Also, is the "dose" of EVs that are applied in this model in any way physiological? For example, how does the dose of Aa EVs compare to the concentration of bacteria (CFUs/ml) used to prepare the EVs?

These issues must addressed before further consideration of the manuscript.

We cannot make any decision about publication until we have seen the revised manuscript and your response to the reviewers' comments. Your revised manuscript is also likely to be sent to reviewers for further evaluation.

Sincerely,

Dana J. Philpott

Academic Editor

PLOS Pathogens

David Skurnik

Section Editor

PLOS Pathogens

Kasturi Haldar

Editor-in-Chief

PLOS Pathogens

orcid.org/0000-0001-5065-158X

Michael Malim

Editor-in-Chief

PLOS Pathogens

orcid.org/0000-0002-7699-2064

Thank you for submitting your work to PLOS Pathogens and I am sorry for the delay getting the reviews back to you. The Reviewers found the work of interest but there are some major issues that need to be addressed with additional experimental evidence. In particular, I agree with Reviewer 1 that specificity of the response needs to be addressed. Do other EVs have the same response as Aa EVs and are they trafficked similarly? These controls need to be demonstrated. Also, is the "dose" of EVs that are applied in this model in any way physiological? For example, how does the dose of Aa EVs compare to the concentration of bacteria (CFUs/ml) used to prepare the EVs?

These issues must addressed before further consideration of the manuscript.

Reviewer's Responses to Questions

**Part I - Summary**

Reviewer #1: This research article regarding periodontal pathogen EV-induced neruoinflammation by Ha et al is an interesting and in-depth study of the ability of extracellular vesicles (EVs) from the pathogen Aggregatibacter actinomycetemcomitans (Aa) introduced via the oral route to induce inflammation in the brain in a mouse model. The authors utilize two administrative routes via the oral cavity (gingival soaking and injection) and investigate neuroinflammatory responses using signaling pathway-deficient (MyD88 KO) and immune-competent (WT) mice. They followed the trafficking of the EVs using fluorescently labeled EVs and in situ imaging. This work follows on two very closely related studies from this group that already demonstrated bEV trafficking and exRNA delivery to the brain (2020, 2019), which reduces the novel aspects of this report. The major critical areas concern dosage and specificity as detailed below. At this point in these studies, it is expected that such questions need to be addressed in order to move the field forward in a significant way.

Reviewer #2: The study suggested that EVs derived from periodontopathogens such as Aa might be involved in pathogenic pathways for neuroinflammatory diseases, neuropathic pain, and other systemic inflammatory symptoms as a comorbidity of periodontitis. It was interesting and novelty.

**Part II – Major Issues: Key Experiments Required for Acceptance**

Reviewer #1: Major comments:

1. A serious question is the matter of physiological dosing. The authors do not address experimentally whether the EV dose they are administering via the ginigival route is at all physiologically reasonable. How does the amount of EVs entering the gingiva compare to the amount produced by Aa? Is it an amount that is absurdly high such that it would never be attained by typical Aa colonization in the gingival tissue? This study claims to test the hypothesis whether periodontitis facilitates the migration of oral bEVs to brain to induce neuroinflammation (lines 252-3), yet the experimental model is not periodontitis, it is application of purified EVs by injection or oral gel. Notably, the authors claim (lines 87-92): "Previously we showed that EVs of Aa and their RNA cargo could cross the blood-brain barrier (BBB) and increase proinflammatory cytokine expression in the mouse brain [20,21]. This was the first direct evidence that blood-borne bEVs can cross the mouse BBB and cause neuroinflammation. However, it remained unclear whether EVs originated from periodontopathogens could enter the brain through leaky gum during PD development." However, the study here does not actually address whether the EVs that are made 'during PD development' traffic in the same manner or with the same immunoinflammatory results as those that have been isolated and introduced in the quantity that they were during these experiments.

2. Another critical question that is not addressed by these experiments is specificity. Obviously, there are numerous commensal species and strains of Gram-negative oral bacteria other than Aa which are also making EVs. Why would those EVs not traffic similarly through the BBB and generate neuroinflammatory disease? Such qualifications should also be considered to modify statements such as (Lines 349-351) "These findings suggest certain inflammatory disorders may be associated with the translocation of bEVs to systemic organs. Therefore, the ability of bEVs to circulate throughout the body may help in understanding autoimmune disorders, up to 50% of which lack a definite etiological cause [73]." There must be much more to generate specificity of transport and overcoming barriers in tissues, otherwise all tissues and sites would be flooded with bEVs that were produced by the human microbiome and all would cause pathologies due to PAMPs carried by the bEVs.

Reviewer #2: 1. Why choose 24h for osteoclastic activity?

2. Line 949-950, After 24 h, gingival tissues were stained for TRAP activity, the representative images of TRAP-positive cells 951 being shown (scale bar: 200 μm). It was unreasonable to analyse TRAP staining in gingival tissues.

**Part III – Minor Issues: Editorial and Data Presentation Modifications**

Reviewer #1: (No Response)

Reviewer #2: 1.Lack of animal ethics approval number.

2.How to exclude the influence of unilateral molding on the opposite side?

3.The image resolution was low.

4.Quantitative statistics are required for positive TRAP staining cells.

5.p value should be in italics.

6.*p≤0.05 was wrong.

7.Scale bar was missing in Fig 6. D.

8.The volume of injected Aa EVs and Aa EV-soaked gel should be demonstrated.

PLOS authors have the option to publish the peer review history of their article (what does this mean?). If published, this will include your full peer review and any attached files.

Reviewer #1: No

Reviewer #2: No
---

## [Decision Letter · Decision Letter 1]

10 Oct 2023

Dear Prof. Lee,

We are pleased to inform you that your manuscript 'Periodontitis promotes bacterial extracellular vesicle-induced neuroinflammation in the brain and trigeminal ganglion' has been provisionally accepted for publication in PLOS Pathogens.

Before your manuscript can be formally accepted you will need to complete some formatting changes, which you will receive in a follow up email. A member of our team will be in touch with a set of requests. Moreover, please add the suggested addition to the transmission electron microscopy methods (more details on staining procedure).

Best regards,

Dana J. Philpott

Academic Editor

PLOS Pathogens

David Skurnik

Section Editor

PLOS Pathogens

Kasturi Haldar

Editor-in-Chief

PLOS Pathogens

orcid.org/0000-0001-5065-158X

Michael Malim

Editor-in-Chief

PLOS Pathogens

orcid.org/0000-0002-7699-2064

Reviewer Comments (if any, and for reference):

Reviewer's Responses to Questions

**Part I - Summary**

Reviewer #1: The authors have addressed the major concerns. A reasonable vesicle dose has been established. Content has been slightly edited to reflect key deficits regarding specificity. No further modifications are needed except the minor modification to the methods-- see below.

**Part II – Major Issues: Key Experiments Required for Acceptance**

Reviewer #1: (No Response)

**Part III – Minor Issues: Editorial and Data Presentation Modifications**

Reviewer #1: There is insufficient detail regarding the Transmission Electron Microscopy methodology. Staining methods are missing.

PLOS authors have the option to publish the peer review history of their article (what does this mean?). If published, this will include your full peer review and any attached files.

Reviewer #1: No

---

## [Editor Report · Acceptance letter]

18 Oct 2023

Dear Prof. Lee,

We are delighted to inform you that your manuscript, "Periodontitis promotes bacterial extracellular vesicle-induced neuroinflammation in the brain and trigeminal ganglion," has been formally accepted for publication in PLOS Pathogens.

Best regards,

Kasturi Haldar

Editor-in-Chief

PLOS Pathogens

orcid.org/0000-0001-5065-158X

Michael Malim

Editor-in-Chief

PLOS Pathogens

orcid.org/0000-0002-7699-2064